# The Role of Indoleamine 2, 3-Dioxygenase 1 in Regulating Tumor Microenvironment

**DOI:** 10.3390/cancers14112756

**Published:** 2022-06-01

**Authors:** Xinting Huang, Feng Zhang, Xiaobo Wang, Ke Liu

**Affiliations:** 1Department of Ophthalmology, The Second Xiangya Hospital of Central South University, Changsha 410011, China; 208211123@csu.edu.cn (X.H.); 218211107@csu.edu.cn (X.W.); 2The Third Xiangya Hospital of Central South University, Changsha 410013, China; amanda_2017@csu.edu.cn

**Keywords:** indoleamine 2,3-dioxygenase 1, tumor microenvironment, interferon-γ, dendritic cell, myeloid-derived suppressor cell, regulatory T cell

## Abstract

**Simple Summary:**

Tumor microenvironment is a complex and dynamically changing entity, which is crucial for tumor development. Indoleamine 2, 3-dioxygenase 1 is elevated in the tumor microenvironment and is strongly associated with tumor histological malignancy. Therefore, the Indoleamine 2, 3-dioxygenase 1 metabolic pathway as a potential tumor immune escape could be used as a novel strategy for cancer therapy. However, the current phase III clinical trials did not achieve a desired result. Thus, it is imperative to further explore the immunosuppressive mechanism mediated by indoleamine 2,3-dioxygenase 1 in the tumor microenvironment to optimize clinical trial treatment strategies.

**Abstract:**

Indoleamine 2, 3-dioxygenase 1 (IDO1) is a rate-limiting enzyme that metabolizes an essential amino acid tryptophan (Trp) into kynurenine (Kyn), and it promotes the occurrence of immunosuppressive effects by regulating the consumption of Trp and the accumulation of Kyn in the tumor microenvironment (TME). Recent studies have shown that the main cellular components of TME interact with each other through this pathway to promote the formation of tumor immunosuppressive microenvironment. Here, we review the role of the immunosuppression mechanisms mediated by the IDO1 pathway in tumor growth. We discuss obstacles encountered in using IDO1 as a new tumor immunotherapy target, as well as the current clinical research progress.

## 1. Introduction

IDO is a cellular metabolic enzyme that metabolizes Trp into Kyn, then binds and activates the aryl hydrocarbon receptor (AhR) [1]. IDO becomes well known for its function as an essential amino acid Trp degradation enzyme in the body. The IDO gene family includes IDO1 and IDO2. However, compared to IDO1, IDO2 is a poor producer of Kyn. Although IDO2 can also initiate the Trp pathway, the affinity and catalytic efficacy of IDO2 for the substrate are very low [2]. Therefore, IDO2 may have little effect on the whole Trp metabolism [3,4,5]. The expression level of IDO1 closely correlates with poor prognosis of tumor, and IDO1 inhibitors have shown significantly limit tumor growth [6,7,8,9,10].

TME is a complex, dynamic entity which composition varies by the tumor type, but primarily includes immune cells that inhibit the antitumor immune response, blood vessels, and extracellular matrix. TME has been extensively implicated in tumorigenesis as it harbors tumor cells that interact with surrounding cells especially immune cells to influence tumor growth, metastasis, and response to therapy [11]. Immune cells in TME mainly include dendritic cells (DCs), regulatory T cells (Tregs), myeloid-derived suppressor cells (MDSCs), nature kill cells (NK cells), and tumor-associated macrophages (TAMs), activating an immunosuppressive environment through a variety of mechanisms [12,13]. Accordingly, the immunosuppressive TME is one of the major barriers to effective antitumor therapy. In recent years, we have witnessed enormous growth in studies on the IDO1-mediated immunosuppressive network in TME. Upon IDO1 catalysis, the metabolite Kyn binds to AhR, regulating DCs towards an immunosuppressive phenotype [14,15,16] and activating immunosuppressive function of MDSCs [17,18,19], then resulting CD4 + T cells differentiate into Tregs. IDO1 affects NK cells by regulating the critical cytotoxic receptors expression [20,21]. The effect of IDO1 on immune cells and other TME-related cells is not independent. IDO1 + DCs promote formation of Tregs and induce Tregs to inhibit normal immune surveillance [22,23,24,25]. Tregs then recruit MDSCs and promote TAMs proliferation to enhance IDO1-drive immunosuppressive network [19,26]. Overall, IDO1-Kyn-AhR pathway plays a vital role in forming immunosuppressive TME and promotes TME transform from normal immunogenic to tolerogenic.

Hence, interfering with the IDO1 pathway targeting cancer has become one of the focused areas in cancer immunotherapy research in recent years. Numerous small-molecule IDO1 inhibitors have been reported in clinical trials. Here, we provide an exhaustive review of the role of the IDO1 metabolic pathway in the constituent cells of TME over the past decades, and depicts the IDO1-mediated immunosuppressive network in TME. This review may raise new ideas for optimizing a novel direction of clinical trial strategies.

## 2. Biology and Essential Role of IDO1

IDO1 is a heme-containing enzyme catalyzing the conversion of Trp to Kyn by cleaving the 2, 3-double bond of the indole ring. Kyn is then converted into other active metabolites through a series of enzymatic reactions, further resulting in the production of nicotinamide adenine dinucleotide + (NAD +) and adenosine triphosphate (ATP) to promote cellular metabolism [27]. IDO1 gene is located on chromosome 8 (8p12-p11 in humans, 8 A2 in mice), and its span is about 15 kb [28,29]. The promoter of the IDO1 gene contains three interferon-activated sites (GAS) and two interferon-stimulated response elements (ISRE), which interact with interferon regulatory factor 1 (IRF1) and signal and activator of transcription (STAT1), respectively [28,29,30]. These interactions make IDO1 responds strongly to interferons (IFNs). In this respect, inflammatory states such as infection or tumorigenesis induce the generation of IFNs and may co-induce the expression of IDO1 in some cells [31]. These IDO1-expressing cells, including MDSCs, DCs, and macrophages, are closely related to the restriction of normal immune response, which is consistent with the high expression of IDO1 in some tumor tissues [7,32,33,34]. Accordingly, inhibition of IFNs production and interference with IDO1 expression will be an inspiring and promising direction in tumor therapy.

## 3. Regulation of IDO1

### 3.1. JAK/STAT Signaling Pathway

IFN-γ is a potent inducer of IDO1 expression, and its transcription induction mechanism mainly depends on Janus kinase (JAK), IRF1, and STAT1 transcription factors [35,36,37,38]. In response to IFN-γ, STAT1 is firstly phosphorylated by JAK, then dimerized and recruited into the nucleus, where it binds to the upstream of IDO1 encoding region GAS-2 and GAS-3 sites and directly activates IDO1 expression. Simultaneously, IFN-γ and STAT1 also indirectly activate IDO1 expression by synergistically inducing the synthesis of IRF-1 and binding to ISRE-1 and ISRE-2 sites [39,40,41,42,43,44]. However, complete STAT1 activation process depends on IFN-γ-induced activation of the PI3Ka pathway [45,46]. Compared with STAT1, a published study has shown that IRF-1 combined with ISRE site is a more critical mechanism for inducing IDO1 gene expression [39]. Furthermore, IFN-β is reported to upregulate IDO1 expression via activating STAT1/STAT2, and activate IDO1-Kyn-AhR metabolic circuitry like IFN-γ signaling [47,48]. Unexpectedly, cell metabolism glycolysis involves the JAK/STAT/IDO1 axis and maintains cellular IDO1 expression in responding to inflammatory cues. During the process of providing fuel for the hexosamine biosynthesis pathway, glycolysis generates GlcNAc, which is utilized by the O-GlcNAc Transferase (OGT) for the o-GlcNAc glycosylation of STAT1 [49]. The STAT1-O-GlcNacylation signaling circuit is necessary for maintaining the activation of STAT1 and its stable downstream effects.

### 3.2. NF-κB Signaling Pathway

NF-κB transcription factors have been implicated in IDO1 induction [43]. NF-κB I (IkB) can be induced by canonical (classical) and noncanonical (alternative) signaling pathways. The canonical pathway involves activation of IkB kinase-β (IKKβ) [50,51], while the noncanonical pathway is strictly dependent on IkB kinase α (IKKα) homodimers [52,53]. IKKα and IKKβ could have the opposite effect, as IKKβ mediates the response to pro-inflammation, and IKKα is implicated in remission of the early inflammatory process, lymphoid organ formation and immune cell maturation [46,54,55,56]. It is noteworthy that IFN-γ requires (IKKα) to activate IDO1 expression [57], which suggest that IDO1 expression requires noncanonical NF-κB pathway to inhibit T cell activity and promote the expansion of immunosuppressive T cells. In current studies, IFN-γ is the primary inducer of IDO1, but other inflammatory stimuli including tumor necrosis factor α (TNF-α) and Interleukin-6 (IL-6), could also induce IDO1. Similarly, TNF-α can synergistically enhance IDO1 transcription through NF-κB relocation and increase the binding of IFN-γ-transactivated factors to GAS and ISRE sites [39,58]. Still, the induction level is markedly less than IFN-γ. In addition, it is interesting that IL-6 plays a dual role in the co-regulation of IDO1 expression. IL-6 negatively regulates IDO1 expression by inducing SOCS3 in normal DCs. Some studies show that blocking the noncanonical NF-κB pathway reduces IL-6 in DCs [46,59]. Opposing effects of IL-6 have been observed in DCs, where a high IDO1 level is detected in Epstein-Barr virus (EBV) infection-induced human monocyte-derived macrophages (MDMs) [60]. EBV infection increases the production of TNF-α and IL-6, which subsequently synergistically upregulates IDO1 expression in MDMs. Similar results have been observed in cancer-related fibroblasts (CAFs) and MDSCs [61,62]. These opposite effects on regulating IDO1 expression in different cell states and cells indicate that IL-6 may mediate two pathways involved in the transcriptional expression process.

In summary (Figure 1), JAK/STAT and noncanonical NF-κB pathways are the major signaling pathways activating IDO1 expression, while a second signal is necessary to activate the enzyme. A published study has shown that Prostaglandin E2 (PGE2) induces the expression of IDO1 via catalyzing the formation of cAMP and activating PKA, and the catalytic activity of IDO1 is then activated through TNF receptor (TNF-R) or a Toll-like receptor (TLR) signal [63]. De novo heme synthesis is required for induction of the active IDO1 enzyme in monocyte-derived macrophages [64]. Cyclooxygenase (COX)-2, the rate-limiting enzyme in the synthesis of prostaglandins, is significantly associated with the IDO1 catalytic activity [65]. Moreover, the metabolite N-acetylserotonin (NAS) of serotonin pathway acts as a positive allosteric modulator of the IDO1 enzyme to directly activate IDO1 [66]. Other pathways, including IL-6-mediated pathways, may each be involved in IDO transcriptional expression processes through different mechanisms, although the detailed mechanisms need to be further explored.

## 4. Role of IDO1 in TME

Tumor cells are closely related to the surrounding environment and interact continuously. In the process of tumor development, TME promotes the proliferation and metastatic spread of cancer cells by coordinating with tumor cells. Considering the vital role of TME in tumorigenesis, an increasing number of studies are focusing on new TME targets in immunotherapy. Intriguingly, the individual components of TME that constitute the tumor immunosuppressive network may be IDO1 expression-dependent. Friberget et al. first reported the IDO1 pathway as a possible tumor immune escape mechanism in 2002 when they observed IDO1 expression by monocytes in tumor tissues and tumor-draining lymph nodes [67]. In the past decade, numerous studies have observed high IDO1 levels in human tumors, including glioblastoma, head and neck squamous cell carcinoma, breast cancer adrenocortical carcinoma, esophageal squamous cell carcinoma, gastric cancer, and colon cancer [68,69,70,71,72,73,74,75].

In addition to the tumor cells, IDO1 is also expressed by immune-related cells. IDO1-expressing plasmacytoid DCs were found in malignant melanoma tumor-draining lymph nodes (TDLNs) [76,77]; IDO-expressing macrophages have also been isolated from human ovarian cancer [78]. The IDO1 level is associated with the histological malignant grade and adverse prognosis. Jiao et al. observed increasing IDO1 expression in TME after neoadjuvant chemoradiotherapy and neoadjuvant chemotherapy accompanied by poor pathological response and prognosis [70]. IDO1 pathway accelerates colorectal cancer growth, local invasion and suppresses CD8 + T cell response. In contrast, IDO1 inhibitors can significantly potentiate Th1 cytokines and myeloid cell-modulating factors in TME and improve radiation therapy outcomes [79,80]. These results suggest that IDO1 has a non-negligible role in TME. IDO1-mediated Trp degradation pathway, resulting in decreased Trp levels and increased Kyn levels. In this respect, low levels of Trp activates the amino acid-sensitive general control nonderepressible 2 (GCN2) pathway and the mammalian target of rapamycin (mTOR) kinase pathway to increase effector T cell deactivation and apoptosis [81,82,83]. Accumulated Kyn binds AhR to induce apoptosis of effector T cells and promote the transformation of naive CD4 T cells to immunosuppressive Forkhead box P3 (Foxp3) Treg cells [1]. Furthermore, IDO1 induces tumor angiogenesis in vivo by regulating the IFN-γ/IL-6 balance [84,85]. These mechanisms may significantly inhibit tumor immunity and promote tumor growth and invasion via the IDO1 pathway.

### 4.1. Immunomodulatory Effects of IDO1 on Tumor-Associated Dendritic Cells

Dendritic cells (DCs) are specialized antigen-presenting cells that coordinate the body’s immune response. DCs play an essential role in maintaining the immunosuppressive state in TME. TME is infiltrated with DCs in different maturation stages and subsets [86]. Immunosuppressive factors such as IFN-γ, PGE2, IL-6, and transforming growth factor-β (TGF-β) are produced in TME, recruiting DCs to TME and converting them from an immunostimulating phenotype in early stage to an immunosuppressive phenotype in the late stage [87,88,89,90]. Immunosuppressive DCs maintained the tumor-resistant state and sustained growth of tumor.

The formation of tolerogenic DCs (tDCs) and their immunosuppressive function is closely related to IDO1. Many studies have found that DCs in mouse spleens and TDLNs exhibit structural and intact IDO expression under basal conditions [24,77,91]. IDO1 in DCs is induced and activated by ligating the B7 ligand through cytotoxic T-lymphocyte-associated protein 4 (CTLA-4), which plays a key role in peripheral tolerance [92,93,94]. In addition to the ligand-binding modality, the cytokine IFN-γ is also involved in the induction process. Activated by IFN-γ, immunogenic DCs (iDCs) gradually transform into IDO1 + tDCs and lose the ability to activate prime CD8 + T cells [90,95,96]. The induction of IFN-γ is transient, but the long-term maintenance of IDO1 expression that endows DCs with the tolerance phenotype depends on the IDO1-Kyn/AhR-IDO1 loop [95]. It is a self-regulated positive feedback pathway, which is first triggered by IDO1 enzyme to form the Kyn/AhR complex. The Kyn/AhR complex endows DCs with the tolerant phenotype. Then, the positive feedback continues to maintain the IDO1 expression in tDCs [95,96,97]. Moreover, 3-hydroxyanthranilic acid (3-HAA), a Trp metabolite produced downstream of Kyn, promotes the binding of the nuclear coactivator 7 (NCoA7) to AhR in DCs and enhances the transcription of Kyn-driven, AhR-dependent genes [98]. Apart from IFN-γ, the Wnt-β-catenin signaling pathway promotes IDO1 + DC tolerance [99,100,101,102,103]. The Wnt receptors Wnt3a and Wnt5a could induce β-catenin binding to the IDO1 promoter in DCs, and the induced IDO1 expression level was higher than IFN-γ [101]. Especially, Wnt5a plays a leading role in inducing and maintaining IDO1 expression, while Wnt3a-mediated IDO1 expression depends on IFN-γ [99,101].

DCs drive the differentiation and proliferation of Tregs and induce Tregs to inhibit normal immune surveillance. It has been observed in several studies that IDO1 + DCs lead to local T-cells deactivation and suppress the host’s anti-tumor T-cells response [22,23,24]. When IDO1 + tDCs were transformed and formed, the immunosuppressive factor TGF-β and Tregs infiltration significantly increased TME [14,15,16,90,102]. These findings led to a prediction that the T-cells suppressive effect of DCs was mainly mediated by activating Tregs proliferation and Tregs suppressive activity. CD4 + CD25 + T cells differentiated into CD4 + CD25 + Foxp3 + Tregs by specific stimulation of degradation product Kyn induced by IDO + DC [24,25]. However, it is not the newly differentiated Tregs that predominantly inhibit T cells. IDO1 + DC-mediated Trp depletion causes pre-existing quiescent CD4 + CD25 + Foxp3 + Tregs to acquire inhibitory activity by activating the complete amino acid-responsive GCN2 pathway in Tregs [77]. This fraction of Tregs is the major participant in the anti-tumor T cell response. Increased IDO1 expression in DCs also slightly enhances the immunosuppressive effect of MSCs on the immune system [15]. IDO1 appears to construct an immunosuppressive network in the TME through DCs.

### 4.2. Immunomodulatory Effects of IDO1 on Tregs

Tregs are necessary to form self-tolerance in the immune system and belong to the immunosuppressive subtype of CD4 + T cells. Additionally, they participate in the process of inhibiting tumor immune response and promoting tumor growth by suppressing the activation and proliferation of cytotoxic CD8 + T cells and effector T cells.

IDO1 drives the activation process of immunosuppressive phenotype Tregs. Foxp3, as a specific immunosuppressive marker, endows Tregs with the ability of immunosuppression. Absent or mutation of Foxp3 will lead to severe autoimmunity in mice and significantly inhibit tumor growth [104]. As widely reported, Foxp3 and IDO1 are co-expressed in a series of tumor tissues [105,106,107,108], suggesting that IDO1 is involved in the differentiation of T cells into Foxp3 Tregs and promotion of tumor growth. IDO1 expression in TME mainly comes from PDC in TDLNs [24,77]. IDO1 does not act directly. Instead, Kyn activates AhR and induces T cells differentiation into Foxp3 + Tregs [1,24]. Inhibition of AhR leads to a decrease in Foxp3 + Tregs [19,109]. IDO1 affects the immunosuppressive function of Foxp3 + Tregs by regulating mTOR2 and Akt signals. Depletable Trp activates low levels amino acid-sensitive GCN2 and leads to inhibition of mTORC2 complex and Akt phosphorylation [110]. Tregs maintain their inhibitory phenotype by keeping low Akt signaling, which is necessary for the normal activation of effector T cells [111]. Moreover, the IDO1-activated GCN2 kinase pathway mediates cell cycle arrest and incompetence induction. In high IDO1-expressing TME, a significant increase of Foxp3 + Tregs was observed, and Foxp3 + Tregs inhibit CD8 + T cells proliferation in vitro [17,19,112]. IDO1 preferentially activates mature, pre-existing Tregs’ inhibitory activity in the co-culture system while inducing newly differentiated Foxp3 + Tregs to acquire less activity [77].

IDO1-activated Tregs develop and infiltrate the tumor microenvironment, suppressing local immune surveillance and promoting tumor development and metastasis. Compared with the B16-F10 melanoma parental cell line (B16WT) mouse model, Tregs isolated from the B16-F10 melanoma overexpressing IDO (B16IDO) mouse model exhibited a higher level of immunosuppressive factors, including vascular endothelial growth factor (VEGF), CTLA-4, IL-10 and more remarkable ability to suppress immune activation of autologous CD8 + T cells [19]. IDO1-activated Tregs cause proliferation of M2-like tumor-associated macrophages (TAMs), which together with Tregs form the Treg-macrophage suppressive axis to promote the formation of TME [19]. Similarly, Tregs also recruit and activate immunosuppressive MDSCs to infiltrate the tumor tissues [17].

### 4.3. Immunomodulatory Effects of IDO1 on MDSCs

MDSCs are a class of immature myeloid cells in different differentiation stages characterized by immunosuppressive activity. Several studies have provided evidence that MDSCs mediate multiple immunosuppressive mechanisms in TME, including but not limited to, promotion of Treg infiltration and up-regulation of immunosuppressive cytokines [113,114,115]. Depletable Trp and accumulative Kyn in TME may also be closely related to the recruitment and activation of MDSCs in tumor tissues [17].

Glutamine metabolism drives the generation, recruitment, and apoptosis of MDSCs in primary and metastatic tumors, while it is highly related to IDO1 expression. A significant decrease in IDO1 expression can be observed in the tumor when inhibiting glutamine metabolism, accompanied by a robust reversal of the Kyn/Trp ratio [26], which led to the prediction that the increased IDO1 expression activates the immunosuppressive effect of MDSCs in TME. IDO1 level in MDSCs is mainly induced by signal transducer and activator of transcription factor 3 (STAT3)-NF-Kβ-IDO1 pathway. Tumor-derived IL-6 phosphorylates STAT3 and stimulates two noncanonical NF-kB subunits, p52 and RelB to form a dimer that directly binds to the promoter of IDO1, then drives the expression of IDO1 in MDSCs [61,116]. An adaptor protein signal, caspase recruitment domain-containing protein 9 (CARD9), which is highly expressed in myeloid cells, initiates the IDO1 expression via the NF-kβ pathway, thus inhibiting the inhibitory function of MDSCs [117,118]. More generally, these data suggest that CARD9-NF-kβ-IDO1 and IL-6-STAT3-NF-kβ-IDO1 are vital tumor immunomodulatory pathways in MDSCs.

IDO1-expressing MDSCs are involved in tumor immunosuppression and immune escape processes. IDO1 is necessary for MDSCs recruitment to tumor tissues, lymph nodes, and spleens for local immunosuppressive functions, and IDO1-expressing MDSCs are a crucial cell population for immunotherapy resistance in a range of tumors [17,119,120,121,122]. IDO1-expressing MDSCs have been shown to accumulate in chronic lymphocytic leukemia (CLL) patients, suppress T cells activity significantly, and induce suppressive Tregs in vitro [123,124]. Correspondingly, when IDO1 was highly activated in MDSCs, the increased infiltration of CD4 + CD25 + FoxP3 + Tregs and the enhancement of immunosuppressive function could be observed in the tumor. Interestingly, as a part of the feedback pathway, IDO1-induced Tregs are involved in the recruitment and activation of MDSCs. When Tregs were absent in the tumor, the ability of the splenic MDSCs to migrate to the tumor was lost entirely [17,33]. In addition, decreased IDO1 expression leads to a reduced ability of MDSCs to induce the production of FoxP3 + Treg [18].

Hence, MDSCs, Tregs and DCs coordinate to form the tumor immunosuppressive environment through the IDO1 pathway (as shown in Figure 2 and Table 1). However, the immunosuppressive activity of MDSCs also affects a wide range of immune cell subsets. MDSCs down-regulates the expression of key surface markers on B cells, including CD80, CD86, CD95, IgM, HLA-DR, and TACI in vitro through IDO1 expression rather than direct cell-contact, and induces B cells apoptosis-related gene (such as BAX, BCL-2, and FAS) upregulated [125]. Accordingly, MDSCs significantly interfere with B cells proliferation and immune function. Furthermore, MDSCs may produce a variety of chemokines, growth factors, and pro-inflammatory factors (such as IL-2, IL-10, IL-6, TNF-α, IFN- γ, VEGF and GM-CSF), which further promote the formation of TME [17].

### 4.4. Immunomodulatory Effects of IDO1 on Natural Killer Cells

Natural killer (NK) cells can kill tumor cells and virus-infected cells without prior sensitization. Based on the role of NK cells in anti-tumor responses, many studies have shown the potential of NK cells in tumor immunotherapy, which makes NK cells the attractive candidate for tumor therapy [126,127,128]. However, the treatment effect of NK cells in solid tumors still faces serious limitations [129,130]. In addition to the difficulty of NK cells penetrating the tumor in site, the continuous interference of TME’s immunosuppressive effect is also a big obstacle in this limitation. In TME, some tumor cells or tumor-associated cells secrete immunosuppressive factors, including TGF-β, IDO, PGE2, IL-6, and IL-10, which prevent NK cells from performing normal cytotoxic killing function [131,132,133]. Significantly, IDO1 enzyme has a leading effect on the anti-tumor responses of NK cells and its interactions with other TME-related cells.

Tumor cells lead to NK cell dysfunction by promoting IDO1 expression. As shown in recent studies, IDO1 is highly expressed in some tumors, and the generated Kyn binds and activates AhR of NK cells, significantly reducing the expressions of natural cytotoxic receptors NK cell p46-related protein (NKp46) and type II integral membrane protein (NKG2D) on the surface of NK cells [20,21]. NKp46 and NKG2D are key activating receptors of NK cells to bind target cells’ ligands and determine the fate of these cells [134]. Kyn does not directly act on NK cell activating receptors, but regulates receptors expression through JAK-STAT signaling pathway. The JAK-STAT signaling pathway is known to be involved in the maturation, survival, and cytotoxicity of NK cells, and most cytokines can activate or block NK cells by this pathway [135,136,137]. STAT1 and STAT3 directly regulate the expression of NKp46 and NKG2D by binding to the promoter regions of these receptors [21]. When Kyn accumulates around NK cells, the number of phosphorylated forms of STAT1 and STAT3 is reduced, thereby down-regulating the level of NKp46 and NKG2D receptors. IDO1 also interferes with the function of NK cells through another axis. IDO1 can down-regulate NKG2D and NKG2DL in NK cells by upregulating miR-18a, a microRNA (miRNA) that significantly promotes tumor cell growth, metastasis and inhibit apoptosis in breast cancer, ovarian cancer, lung cancer, and hepatocellular carcinoma [138]. The down-regulation of these cytotoxic receptors undoubtedly leads to the dysfunction of NK cells, which in turn affects the interaction between NK cells and other immunosuppressive cells in TME.

NK cells dysfunction mediated by IDO1 impedes the body’s normal anti-tumor immune response. In this respect, recent studies have shown that NKG2DL is expressed in mononuclear subsets of MDSCs derived from subcutaneous lymphoma in mice, suggesting that NK cells play the role of lysing MDSCs by binding NKG2DL [139]. Due to the interference of IDO1, the effect of NK cells on killing immunosuppressive cells in TME was weakened. In addition to being a cell killer, NK cells also secrete IFN-γ to induce IDO1 expression [140]. It is tempting to speculate that while IDO1 affects the anti-tumor effect of NK cells in TME, it may also form an IDO1-NK cells/IFN-γ-IDO1 loop that can promote the increase of self-expression. However, if correct, the combination of IDO1 inhibitors and NK cells immunotherapy would have a more attractive prospect. IDO1 and NK cells may have more interaction than previous thought, and further study is needed to clarify the relationship.

### 4.5. Immunomodulatory Effects of IDO1 on Tumor-Associated Macrophages

TAMs are the most abundant cell type in solid tumors. TAMs usually exhibit an M2-like phenotype to participate in tumor immunosuppression and lead to immune escape of cancer cells via different mechanisms [141,142]. In TME, TAMs and some tumor-derived vascular growth factors such as VEGF, PDGF, CCL2, and CCL8 promote tumor neovascularization and maintain the tumor growth vascular network [143,144,145]. In addition, TAMs promote tumor growth by producing matrix metalloproteinases (MMPs) and cathepsin to degrade the basement membrane [146,147,148]. A number of studies have also provided evidence that TAM seriously interferes with the composition of immune cells in TME [142,149]. While decreasing tumor immune cells, TAMS increased the number of immunosuppressive cells to accelerate the formation of TME. However, which factors promote M2 polarization of TAMs in vivo and how TAMs inhibit anti-tumor immunity in TME remain largely undefined.

TAMs may play a dual role in tumor immunity through the IDO1-Kyn-AhR pathway. In a vitro experiment, IFN-γ was observed to promote autophagy in cervical cancer cells and enhance macrophages phagocytize autophagy-active cancer cells, the mechanism of which may be related to the overexpression of IDO1 and the accumulation of Kyn [150]. This data suggests that IFN-γ may inhibit tumor growth through the IDO1-Kyn-autophagy pathway. However, opposing effects of IDO1 in TAMs have been observed, where IDO1-expressing cancer cells drive TAMs to impair T cell response. In glioblastoma, Kyn produced by cancer cells triggers AhR-signaling to regulate the phenotype of TAMs, and causes the production of adenosine by CD39 and CD73 in TAMs, ultimately leading to CD8 + T cells dysfunction [151]. In addition, AhR signaling mediates tumor immunosuppression by promoting CD155 expression on TAMs [152]. CD155, an immune checkpoint as a target for tumor immunotherapy, is universally expressed in solid tumors [153,154]. Constantly active AhR signaling induces expression of CD155 on TAMs. Inhibiting AhR mitigated CD155 expression on TAMs could reverse tumor immunosuppression. Similarly, the expression level of AhR in TAMs is regulated by IDO1 in tumor. However, IDO1 does not directly control the AhR expression, but through the recruitment of positive regulators STAT1 and STAT3 [151]. TAMs play a role in inhibiting tumor growth and promoting the inhibition of anti-tumor immunity through AhR signaling. Moreover, which of these two opposite effects is dominant in TME remains unclear. Therefore, further studies are still needed to elucidate the association between the IDO1-Kyn-AhR pathway and TAMs in tumors. Targeting AhR of TAMs in tumor immunotherapy may be a novel research direction.

### 4.6. Immunomodulatory Effects of IDO1 on Other TME-Related Cells

Currently, the research of immunotherapy for TME is progressing rapidly. Other than Tregs, MDSCs, TADCs, NK cells, and TAMs, CAFs, B regulatory cells (Bregs), tumor endothelial cells (TECs), and tumor-repopulating cells (TRCs) belong to the cellular components of the tumor immune microenvironment too (Figure 3 and Table 1). However, to our knowledge, no study to date has clearly analyzed the potential relation between IDO1 and these cells.

CAFs originate from normal fibroblasts stimulated by hypoxia in TME or signals from adjacent tumor cells, and are one of the major stromal cells in tumors [155,156]. A recent study by Cui et al. highlighted that interstitial CAFs and microvascular endothelial cells co-express IDO1 in esophageal cancer, which may be an of the mechanisms that cancer cells inhibit host anti-tumor immunity [157]. However, it seems that CAFs recruit other immune cells to express IDO1 in a more indirect way to inhibit T cell response. By contrast, in hepatocellular carcinoma, CAFs secrete IL-6 to mediate STAT3 activation, convert normal DCs into cells with high IDO1 expression and educate them to acquire a tolerogenic phenotype [62]. The relation between CAFs and IDO1 in forming the tumor immune microenvironment remains further studied and explored.

Tumor angiogenesis is an essential condition for tumor progression and metastasis. Rapid tumor growth often leads to increased oxygen consumption, which produces a large number of pro-angiogenic factors in TME and forms a tumor vascular network. IDO1 may be involved in tumor angiogenesis. High microvascular density and worse prognosis of breast cancer is closely related to the expression of IDO1 [158]. Silencing IDO1 gene or using molecule inhibitor of IDO1 significantly reduces the number of tumor neovascularization and inhibits the invasion and migration of cancer cells [159,160]. This inhibition effect of tumor angiogenesis is potentially associated with IDO1 generation by TECs [159]. Accordingly, TECs express IDO1 to accelerate tumor neovascularization and promote tumor growth. However, few studies on the correlation between IDO1 and tumor angiogenesis, and this regulatory effect’s specific signaling transduction pathway remains unclear.

More recently, immunologic tumor dormancy, an immune surveillance escape program against hard microenvironments, has been observed in clinical post-transplantation and surgical remission patients [161,162,163,164]. Immunologic tumor dormancy may be a potential barrier for current immunotherapies. As shown in recent studies, IFN-γ and IFN-β induce TRCs into dormancy by activating IDO1-Kyn-AhR-dependent kinase inhibitor 1B (p27) signaling [47,48,165], which prevents STAT1 and STAT3 signaling to suppressing the cell death and apoptosis. Compared with IFN-γ, IFN-β appears to have a stronger ability to induce TRC dormancy. IFN-β promotes STAT3 serine phosphorylation by JAK1/TyK2 pathway. STAT3 translocation into the nucleus and further activating p27 expression [47]. These data suggest that IFN-γ/IFN-β-induce IDO1-Kyn-AhR pathway involves in the immunologic tumor dormancy and further contributes to immune escape of tumor cells.

IDO1 pathway induces innate immune B cells to be effective induced B regulatory cells (iBregs) [166]. Activated B cells produce IDO1 and TGF-β in a CTLA-4-dependent manner to transform themselves into iBregs, which gain the ability to regulate T cells and drive the generation of Foxp3 + CD4 + T cells [166]. This transformation may accelerate the formation of the immunosuppressive microenvironment, which is conducive to tumor growth. Further studies are needed to shed some light on the interaction between IDO1 and B cells.

## 5. Current Status of IDO1 and TME in Treatment of Cancer, Future Perspective

IDO1 is highly expressed in most tumor tissues and mediates immune escape of tumor cells, so 11 inhibitor drugs targeting IDO1 pathway have entered clinical trials [167]. Over the past few years, clinical trial treatment strategies have mainly included IDO1 inhibitor monotherapy, IDO1 inhibitors combined with checkpoint inhibitors, and IDO1 combined with chemotherapy and radiotherapy (as shown in Table 2). Although several IDO1 inhibitor monotherapies, including indoximod and epacadostat, have entered phase II trials, it is disappointing that no significant tumor reduction and T cell count changes were observed in preclinical work [168,169,170]. The immune checkpoint PD-1/PD-L1 is co-expressed with IDO1 in a range of tumors [171,172,173,174], and anti-PD-1/PD-L1 and anti-CTLA-4 treatments induce IDO1 expression [175,176], suggesting a possible synergistic relationship may exist between the three. Therefore, an increasing number of clinical trials are now focusing on IDO1 inhibitor combination therapies. In this regard, the combination of epacadostat with the immune checkpoint inhibitor pembrolizumab showed encouraging results, promoting the combination therapy into phase III clinical trials [177]. However, this combination therapy did not improve clinical response in phase III trials [178]. The use of triple therapy, combining an IDO1 inhibitor with anti-PD-1/PD-L1 and anti-CTLA-4, showed a reduction in Treg infiltration in vivo, but no difference in survival compared to the CTLA-4/PD-L1 double therapy. The effectiveness of IDO1 inhibitors as a strategy to enhance the therapeutic activity of PD-1/PD-L1 remains unclear. The clinical results of these IDO1 inhibitors have frustrated the research community, but have thus inspired multifactorial thinking and discussion. First, the design details of clinical trials, including inhibitor selection, inhibitor dose, inhibitor combination regimen, cancer type selection, and the patient stratification based on IDO1 expression are all critical in determining the effectiveness of clinical trials [179]. When IDO1 is inhibited, IDO2 and Tryptophan-2, 3-Dioxygenase (TDO) may be potential surrogate pathways for tumors, failing IDO1 inhibitors to inhibit Trp degradation [6,179]. As shown in a recent study, interleukin-4-induced-1 (IL4I1) correlates more with AhR activity than IDO1 or TDO in 32 tumors. IL4I1 activates AhR by producing Kyn, which may also explain the failure of clinical trials using IDO inhibitors [180]. Furthermore, IDO1 inhibitors combined with other types of therapy may induce IDO1 expression up-regulation through various mechanisms. Therefore, exploring the combination of dual IDO/TDO inhibition and TME downstream signaling blockade may become a promising strategic option in future clinical trials.

## 6. Conclusions

This review elucidates the network of immunosuppressive mechanisms regulated by IDO1 in TME, which is involved in the process of tumor development. IDO1 plays a crucial role in tumor immunosuppression, and IDO1 may become a very attractive target for future anti-tumor molecular targeted therapy. However, many unanswered questions remain to be explored before the full potential of IDO in cancer immunotherapy can be harnessed.

**Table 1 cancers-14-02756-t001:** IDO1 affects immune-related cells under different conditions.

Cell Type	Mechanism (Pathway)	Effects	Condition
DCs	IDO1-Kyn-AhR pathway	Endowed with the tolerance phenotype and lose the ability to activate prime CD8 + T cells	Low metastatic lung alveolar carcinoma in vivo [90], mammary carcinoma in vivo [95], and non-tumor in vivo [96]
	IDO1-Kyn-AhR pathway	Promote formation of Foxp3 + Tregs and induce Foxp3 + Tregs to inhibit normal immune surveillance.	Acute myeloid leukemia in vitro [22], and hepatocellular carcinoma in vitro [23], non-tumor in vitro [24,25], low metastatic lung alveolar carcinoma in vivo [90], acute myeloid leukemia in vitro [102], mesothelioma and lung cancer in vivo [16], mammary carcinoma, lung alveolar carcinoma, and colon carcinoma in vivo [14], and non-tumor in vivo [15]
	IDO1-Kyn-AhR pathway and amino acid-responsive GCN2 pathway	Activate pre-existing Tregs suppressive activity	Melanoma in vivo [77]
Tregs	IDO1-Kyn-AhR pathway	Differentiated into CD4 + CD25 + Foxp3 + Tregs	Non-tumor in vitro [1,24,109], melanoma, and colon cancer in vivo [19]
	IDO1-Kyn-AhR pathway and amino acid-responsive GCN2 pathway	Acquire suppressive ability	Melanoma in vivo [77,110]
	IDO1-Kyn-AhR pathway	Inhibit CD8 + T cells proliferation	Melanoma and colon cancer in vivo [19], and Melanoma in vivo [17]
	IDO1-Kyn-AhR pathway	Promote expression of immunosuppressive factors	Melanoma and colon cancer in vivo [19]
	IDO1-Kyn-AhR pathway	Promote TAMs proliferation	Melanoma and colon cancer in vitro [19]
	IDO1-Kyn-AhR pathway	Recruit MDSCs to tumor tissues	Melanoma in vivo [17]
MDSCs	IDO1-Kyn-AhR pathway	Suppress T cells activity and induce suppressive Tregs	Chronic lymphocytic leukemia in vitro [123,124]
	IDO1-Kyn-AhR pathway	Enhance Tregs suppressive activity	Chronic lymphocytic leukemia in vitro [123,124]
	IDO1-Kyn-AhR pathway	Promote formation of Foxp3 + Tregs	Non-tumor in vitro [18]
	IDO1-Kyn-AhR pathway	Interfere with B cells proliferation and immune function	Non-tumor in vitro [125]
	IDO1-Kyn-AhR pathway	Accelerated tumor outgrowth	Melanoma in vivo [17,119], lung cancer in vivo [120], lewis lung carcinoma in vivo [121], and triple-negative breast cancer in vitro [122]
NK cells	IDO1-Kyn-AhR pathway, JAK-STAT pathway, and IDO1-miR-18a-NKG2D-NKG2DL axis	Cytotoxic killing ability dysfunction	Thyroid cancer in vitro [21], pancreatic cancer in vitro [20], and breast cancer in vitro [138]
TAMs	IDO1-Kyn-autophagy pathway	Phagocytic cancer cells with active autophagy	Cervical cancer in vitro [150]
	IDO1-Kyn-AhR pathway, CD39-CD73-adenosine pathway, and IDO1-Kyn-AhR-CD155 pathway	Impair T cell response	Glioblastoma in vivo [151], non-tumor in vitro and breast cancer in vivo [152]
CAFs	IL-6-STAT3-IDO1 pathway	Educate DCs to acquire an IDO1-express tolerogenic phenotype	Hepatocellular carcinoma in vitro [62]
TECs	IDO1-Kyn-AhR pathway	Regulate tumor neovascularization	Clear cell renal cell carcinoma specimen [159], and lung cancer in vivo [160]
TRCs	IDO1-Kyn-AhR-p27 pathway	Enter dormancy	Colon cancer, hepatocellular carcinoma, breast cancer, stomach cancer, and liver cancer in vitro, and melanoma both in vitro and in vivo [47,165]
B cells	IDO1-Kyn-AhR pathway	Transform into iBreg to regulate T cells and drive the generation of Foxp3 + CD4 + T cells	Non-tumor in vitro [166]

**Table 2 cancers-14-02756-t002:** IDO1 inhibitors as a single agent and in combination with other therapies in completed clinical trials.

Agent	Strategy	NCT Number	Phase	Conditions	Clinical Efficacy
Indoximod (1-D-MT)	Single agent	NCT03852446	Early I	Healthy	Unknown
		NCT03372239	I	Healthy	Unknown
		NCT00567931	I	Unspecified adult solid tumor	Unknown
	Sipuleucel-T	NCT01560923	II	Metastatic prostate cancer	Stabel disease (SD) is 50%
	Idarubicin and cytarabine	NCT02835729	I	Acute myeloid leukemia	Unknown
	Temozolomide, cyclophosphamide, etoposide, and radiation	NCT02502708	I	Glioblastoma multiforme, glioma, gliosarcoma, malignant brain tumor, ependymoma, medulloblastoma, diffuse intrinsic pontine glioma, and primary CNS tumor	Unknown
	Nab-Paclitaxel and gemcitabine	NCT02077881	I/II	Metastatic pancreatic adenocarcinoma and metastatic pancreatic cancer	Unknown
	Ipilimumab, nivolumab, and pembrolizumab	NCT02073123	I/II	Metastatic melanoma and stage III-IV melanoma	Unknown
	Docetaxel, indoximod, and paclitaxel	NCT01792050	II	Metastatic breast cancer	Objective response rate (ORR) is 40% and 37%, respectively (indoximod vs. placebo) [181]
	Temozolomide, bevacizumab, and radiation	NCT02052648	I/II	Glioblastoma multiforme, glioma, gliosarcoma, and malignant brain tumor	Unknown
	Docetaxel	NCT01191216	I	Unspecified adult solid tumor	Unknown
Epacadostat (INCB024360)	Single agent	NCT01195311	II	Solid tumors and hematologic malignancy	SD lasting ≥16 weeks was observed in 7 of 52 patients [169]
		NCT01822691	II	Myelodysplastic syndromes	SD in 12 (80%) patients and progressive disease in 3 (20%) patients [170]
	Pembrolizumab	NCT03322540	II	Lung cancer	ORR 32.5%
		NCT03291054	II	Gastrointestinal stromal tumors	Unknown
		NCT02364076	II	Thymic carcinoma, thymus neoplasms, and thymus cancer	SD 52.5%
		NCT03196232	II	Gastric adenocarcinoma, gastroesophageal junction adenocarcinoma, recurrent esophageal carcinoma, recurrent gastric carcinoma, stage IV esophageal cancer AJCC v7, stage IV gastric cancer AJCC v7, and unresectable esophageal carcinoma	Unknown
		NCT02752074	III	Melanoma	No significant differences were found between the treatment groups for progression-free survival [178]
		NCT03361865	III	Urothelial cancer	ORR 31.8%
		NCT03374488	III	Urothelial cancer	ORR 21.4%
	Pembrolizumab and chemotherapy	NCT02862457	I	Neoplasms, carcinoma, and non-small-cell lung	Unknown
		NCT03322566	II	Lung cancer	ORR 26.4%
	Pembrolizumab, oxaliplatin, leucovorin, 5-fluorouracil, gemcitabine, nab-paclitaxel, carboplatin, paclitaxel, pemetrexed, cyclophosphamide, and cisplatin	NCT03085914	I/II	Solid tumor	Partial response (PR): Epa + Pembrolizumab + mFOLFOX6: 55.6%; Epa + Pembrolizumab + 5-FU and Platinum Agent: 45.5%
	Durvalumab (MEDI4736)	NCT02318277	I/II	Solid tumors, head and neck cancer, lung cancer, and urothelial cancer	ORR 12.9% (phase II)
	SD-101 and radiation	NCT03322384	I/II	Advanced solid tumors, lymphoma	Unknown
	MK-3475	NCT02178722	I/II	Microsatellite-instability high colorectal cancer, endometrial cancer, head and neck cancer, hepatocellular carcinoma, gastric cancer, lung cancer, lymphoma, renal cell carcinoma, ovarian cancer, solid tumors, urothelial cancer, breast cancer, and melanoma	ORR: microsatellite-instability high colorectal cancer: 43.8%; melanoma: 60.5%; non-small cell lung cancer: 30.8%; renal cell carcinoma: 32.4%; squamous cell carcinoma of the head and neck: 33.3%; transitional carcinoma of the genitourinary tract: 30.6%
	Nivolumab and chemotherapy	NCT02327078	I/II	B-cell malignancies, colorectal cancer (CRC), head and neck cancer, lung cancer, lymphoma, melanoma, ovarian cancer, and glioblastoma	Unknown
	MELITAC 12.1 Peptide Vaccine	NCT01961115	II	Stage III–IV melanoma	Unknown
	Fludarabine, cyclophosphamide, NK cells, and IL-2	NCT02118285	I	Ovarian cancer, fallopian tube carcinoma, and primary peritoneal carcinoma	Unknown
	DEC-205, NY-ESO-1 Fusion Protein CDX-1401, and Poly ICLC	NCT02166905	I/II	Fallopian tube carcinoma, ovarian carcinoma, and primary peritoneal carcinoma	Unknown
BMS-986205	Single agent	NCT03378310	I	Healthy	Unknown
		NCT03312426	I	Healthy	Unknown
		NCT03374228	I	Healthy	Unknown
		NCT03362411	I	Healthy	Unknown
		NCT03247283	I	Cancer	Unknown
	Nivolumab	NCT03192943	I	Advanced cancer	Unknown
		NCT03792750	I/II	Advanced cancer	Unknown
		NCT03329846	III	Melanoma and skin Cancer	Unknown
	Omeprazole	NCT03936374	I	Healthy	Unknown
	Itraconazole and rifampin	NCT03346837	I	Malignancies multiple	Unknown
navoximod (GDC-0919/NLG919)	Single agent	NCT02048709	I	Solid tumor	(8) 36% had stable disease and (10) 46% had progressive disease [168]
	Atezolizumab	NCT02471846	I	Solid tumor	(6) 9% dose escalation patients achieved PR, (10) 11% expansion patients achieved PR or CR [182]
NLG802	Single agent	NCT03164603	I	Solid tumor	Unknown
SHR9146 (HTI-1090)	Single agent	NCT03208959	I	Advanced solid tumor	Unknown
	Mogamulizumab	NCT02867007	I	Solid tumor, cancer, and carcinoma	Unknown

## Figures and Tables

**Figure 1 cancers-14-02756-f001:**
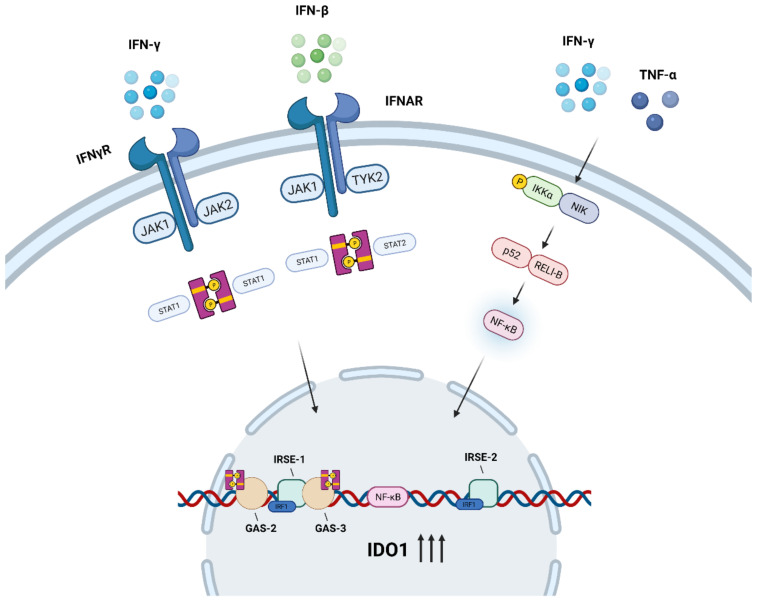
Regulation of activating IDO1 transcription. IFN-γ promotes STAT1 phosphorylation by JAK, then STAT1 dimerization binding to GAS-2 and GAS-3 sites upstream of the IDO1 coding region to activate IDO1 transcription and induce IRF-1 synthesis. IRF-1 binds to ISRE-1 and ISRE-2 sites in concert with STAT1 to promote IDO expression. IFN-β upregulates IDO1 expression via activating JAK1/TyK2 and STAT1/STAT2. Another regulatory pathway, activation of IKKα by NF-κB-inducing kinase (NIK) results in the formation of p52-REL-B dimers, which promotes NF-κB translocation and attaches to the IDO1 coding region. TNF-α synergistically enhancing the IDO1 induction effect of IFN-γ.

**Figure 2 cancers-14-02756-f002:**
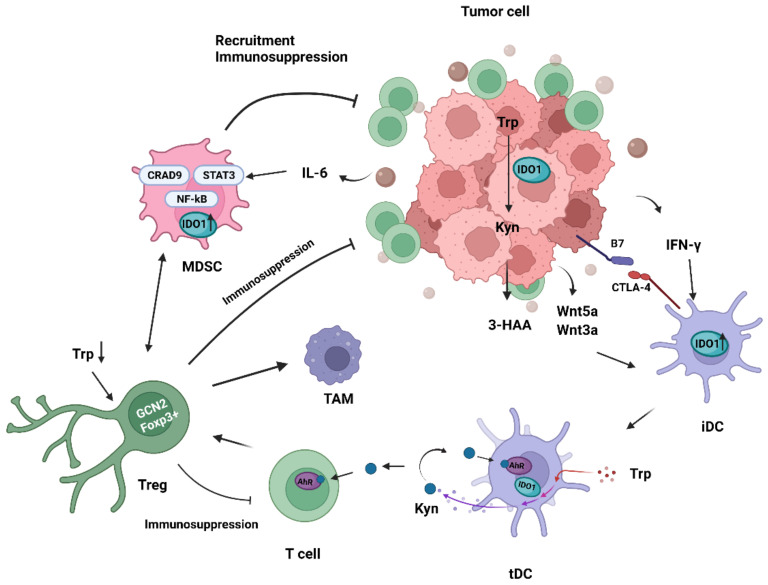
Immunosuppression network mechanism of TADCs, Tregs, and MDSCs in TME. iDCs activate IDO1 expression through CTLA-4 ligation with B7, then are transformed into tDCs lacking the function of activating CD8 + T-cell upon induction of IFN-γ from TME. IDO1 of tDCs can be strongly induced by the Kyn metabolite 3-HAA and the Wnt3a and Wnt5a receptors accumulated in TME. Kyn produced by tDCs metabolism binds to AhR of T cells, stimulating their differentiation into Foxp3 + Tregs. In response to depleted Trp, the GCN2 pathway of Foxp3 + Tregs is activated to suppress tumor immune responses. Tumor-derived IL-6 promotes STAT3 phosphorylation, which upregulates NF-κB-driven IDO1 expression and activates immunosuppressive functions of MDSCs. On the other hand, high expression of CARD9 in MDSCs inhibits immunosuppressive function via the NF-κB pathway. Foxp3 + Tregs and MDSCs interact to influence each other’s ability to migrate to tumors. Moreover, IDO1-activated Tregs cause proliferation of TAMs.

**Figure 3 cancers-14-02756-f003:**
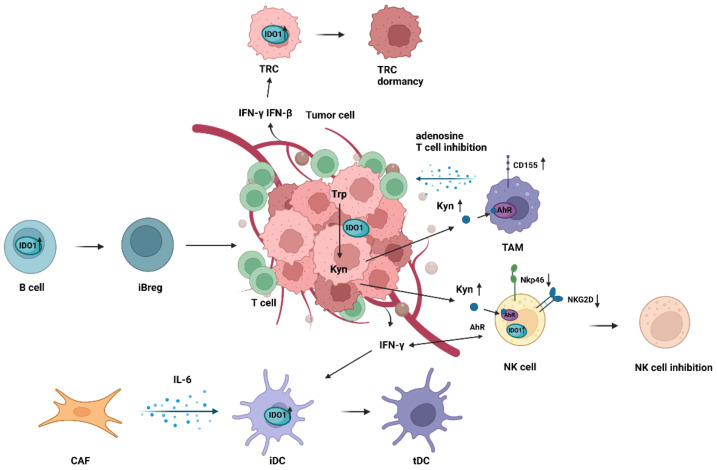
Immunosuppression mechanism of other TME-related cells in TME. IDO1 metabolism in tumors produces an overaccumulation of Kyn. Kyn binds and activates AhR of TAMs, drives TAMs to secrete adenosine to interfere with T cells’ immune function in TME. CD155 expression on TAMs also is upregulated to promote tumor immunosuppression. Kyn binds and activates AhR of NK cells and downregulates NKp46 and NKG2D receptor expression. The significant reduction of these natural cytotoxic receptors inhibits the function of NK cells to kill tumor cells. The major tumor stromal cells CAFs secrete IL-6, which educates iDCs to acquire a tolerogenic phenotype, and further promotes tumor immunosuppression. IFN-γ/IFN-β from tumor induce TRCs into dormancy by activating IDO1 expression. Increased IDO1 expression in B cells drives their conversion to iBregs. iBregs are involved in regulating the immune function of T cells and Tregs in TME.

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
