# Peer review of "The Role of Indoleamine 2, 3-Dioxygenase 1 in Regulating Tumor Microenvironment"

_cancers, 2022, doi:10.3390/cancers14112756_

Round 1

Reviewer 1 Report

Dr  Xinting Huang and Collegues provide a comprehensive review article focusing on  Indoleamine 2, 3-dioxygenase 1 (IDO1), as responsible for defining an immunosuppressive microenvironment. Authors have reviewied the IDO1-dependent immunosuppression; and discussed the status of IDO1 as an immunotherapy target, considering both obstacles and  clinical research progresses. It is an interesting review article, well written. It could be strengthen by addressing the following points.

1) When referring to NF-kB, it would be useful to differentiate between canonical- and non canonical-NF-KB and refer published studies to this distinction (if this is available from the Literature).

2) The review provides a clear biological significance for the role of IDO1 in mediating an immunosuppressive microenvironment. Authors should better locate the knowledge on IDO1/immunosuppressive milieu and the specific tumor context. For example, Authors refer to “tumor”, “tumor types”, without specifying which context. For example, adding a table could be of help.

3) Whenever possible, the positive and negative results from IDO1-based trials should be included and detailed using a table format.

Reviewer 2 Report

In this review, the authors attempt to describe the role of IDO in the tumor microenvironment. They focus on signaling pathways that are involved in IDO induction in different cell types related to tumor environment and the potential effect of IDO induction in this microenvironment.

There are many inaccuracies in this review as well as lack of precision as follows:

-line 37 the authors can’t say soluble factor for Kyn

-line 57 responds (with s)

-lines 58/59 the sentence needs to be rewritten

-lines 76/77 the sentence isn’t clear, “cell metabolism” is too broad

-lines 104-107 they can’t compare signaling pathways such as JAK/STAT to cytokine IL-6

-the paragraph related to the induction of IDO is too focused on transcription of IDO, other mechanisms that modify IDO activity exist. These mechanisms should be at least mentioned

-line 149 maintain (without e)

-line 176, they should define Tregs

-lines 192-212 there are many redundancies. They should better organize the paragraph

-line 269 they can’t say NK cells are as lymphocytes

-lines 282-283 “ Kyn enters the NK cells by combining with AHR” . Do the authors mean Kyn binds AHR?

-line 287 Kyn regulates “WHAT”. A word is missing here

-line 292 how Kyn accumulates around NK cells as the authors said, this changes the phosphorylation of STAT1/3 ?

-326 why TAMs play a dual role through IDO ?

-330-331 I don’t understand why “IFNg may inhibit tumor growth through kyn is at the opposite of IDO-expressing cancer cells impair T cell response” ? if there is T cell impairment, this may favor tumor growth  

-line 335 “AHR signaling mediates tumor immunosuppression by regulating CD115 on TAM” why ? and what is CD115 and its function

-there is an overuse of the term “regulation”, we don’t know whether it is in positive or negative sense

-356 IDO is not produced!

-383 “IDO1 tryptophan degradation” has no sense

-the fail of clinical trials using IDO inhibitor could be also due to IL4I1 (see Sadik et al., 2020, Cell 182, 1252–1270)

-fig 2 AHR is a transcriptional factor, it is not a receptor located on cell membrane!

Reviewer 3 Report

In general, indoleamine 2,3-dioxygenase 1 (IDO1) was discovered more than 50 years, and 20 years ago, tryptophan catabolism was discovered. Since then, much work has been focused on the IDO1. The authors summarized how IDO1 triggers the formation of a tolerogenic tumor microenvironment via affecting its main cellular components and corresponding pathways that promote immune tolerance. The review is well written and readable; however, how it highlights the differences when compared to other review papers on this topic is important. Hence, some suggestions are listed below for your consideration.

  1. The introduction of IDO could be expanded to describe the progress in the area and highlight the importance of your topic. The current introduction cannot tell the differences compared with other review papers on the subject of IDO.
  2. The author mentioned IDO2, i.e. line 34, “But compared to IDO1, IDO2 is less focused, and its role in 34 cancer remains unknown.” However, the description is weak, and it might be more detailed, and references should be added.
  3. Page 37-42. It would be better if it were more closely related to IDO. For example, Kyn regulates the function of many cells, but what is the connection between those cells and IDO1? If there is no connection, how can one conclude that “IDO1 plays a vital role in forming immunosuppressive TME”? The introductory section should be more detailed for better understanding, especially for a general introduction to this research field, what the highlight of your review compared to the published one.
  4. Section 3 and 4. One table might be added to organize the research progress in each subtitle.
  5. Figure 2. I wonder if the TAM and NK mentioned in section 4 are related to this figure? If yes, should add them in?
  6. In addition, activated IFN-b is also reported to lead to TRC dormancy, by either initiating Jak1/Tyk signalling for IDO1-independent p27 upregulation, or activating IDO1 and sharing the IDO1/Kyn/AhR metabolic circuitry with IFN-g signalling. It might be complementary to the pathway in the review. Besides, some more recent research and review papers should be included in the study.
  7. The title and subtitles could be more details to reflect the information the authors want to present. For example, “IDO1 and TME,” “IDO1 and MDSCs”, etc. It could be one short sentence instead of two words.
  8. Section 4. Is there any logic related to the ordering or any other better ways to arrange this section?

Round 2

Reviewer 3 Report

Thanks for the authors’ revision. There are some minors for your consideration.

  1. The introduction could be enhanced further to reflect the importance of your review.
  2. Page 281, 373, and 423. The authors provide the descriptions of table 1 and table 2, while I did not find they are included in the manuscript.
